# A German Validation of Four Questionnaires Crucial to the Study of Time Perception: BPS, CFC-14, SAQ, and MQT

**DOI:** 10.3390/ijerph17228477

**Published:** 2020-11-16

**Authors:** Sebastian L. Kübel, Marc Wittmann

**Affiliations:** 1Institute for Frontier Areas of Psychology and Mental Health, 79098 Freiburg im Breisgau, Germany; s.kuebel@csl.mpg.de; 2Department of Criminology, Max Planck Institute for the Study of Crime, Security and Law, 79100 Freiburg im Breisgau, Germany

**Keywords:** validation, time perception, boredom proneness, time perspective, interoceptive awareness, metacognition, impulsivity, self-control

## Abstract

We aimed to validate four established questionnaires related to time perception in German (Consideration of Future Consequences-14 scale (CFC-14), Boredom Proneness Scale (BPS), Metacognitive Questionnaire on Time (MQT), and Self-Awareness Questionnaire (SAQ)) using a back-translation method. Exploratory factor analyses were conducted on the data of 273 German-speaking participants to evaluate the factorial structures. Internal consistencies indicated good reliability values of the questionnaires and the respective subscales, except for the MQT. Intercorrelations between the questionnaires were examined to test their external validities and gain additional insight into the associations among the constructs. The consideration of future consequences was negatively linked to boredom proneness, whereas interoceptive awareness of one’s bodily sensations was positively associated with boredom proneness. Additionally, interoceptive awareness was linked to metacognitive beliefs about which factors influence time perception. The results are discussed in regard to human time perception. Conclusion: The validated German questionnaires can now be used in research projects. Initial observations on how the questionnaires are related to each other fit the current knowledge on how human time perception works, yielding the first evidence for the external validity of the German versions of these established questionnaires. For evidence of criterion validity, future studies should more thoroughly investigate the external validities analyzing the correlations with other validated measures.

## 1. Introduction

Boredom, consideration of future consequences, metacognition of time, and self-awareness at first sight do not seem to have much in common. A closer look at these constructs shows that they do share a common ground—time perception. In the following paragraphs, each of the constructs will be briefly described based on empirical findings. We demonstrate that thoughts about how human time perception works influence judgments of duration. We show that awareness of our bodily sensations fundamentally shapes how time is perceived. We explain that the frequency of feeling bored and the extent to which temporally immediate or more remote consequences are considered in decision-making are driven by how the passage of time is perceived. When we are bored, we perceive ourselves in a seemingly endless present moment, like in waiting situations [1]. Individuals are more likely to consider future consequences when the future event feels less distant [2]. More impulsive persons consider future consequences less (a phenomenon also referred to as temporal myopia); what matters are immediate rewards. Highly impulsive persons tend to overestimate durations; they perceive the passage of time as relatively slower [3,4]. Potential future consequences are not in their temporal focus when making decisions.

As these findings indicate, all the constructs are linked to time perception; we, therefore, aimed to validate the German versions of established questionnaires assessing these constructs (Boredom Proneness Scale (BPS), Consideration of Future Consequences-14 scale (CFC-14), Self-Awareness Questionnaire (SAQ), and Metacognitive Questionnaire on Time (MQT)). For this purpose, exploratory factor analyses (EFA) were conducted on the translated versions of these scales and internal consistencies were calculated. Intercorrelations of the German questionnaires were computed to assess the relationships between them, and the results are discussed concerning theories and previous empirical research on these topics.

In the following sections, we introduce each construct assessed by the questionnaires in a tripartite structure: first, we provide a brief overview of the theoretical scientific state of knowledge and its relation to human time perception; second, we concisely report associations to other constructs found in previous studies with the particular questionnaire in its original language; third, in case they exist, we present alternative or similar measures for the respective construct and demonstrate what makes the scales validated here unique. Detailed information on the questionnaires can then be found in the methods section.

### 1.1. Boredom

Boredom can be defined as a transient experience of low levels of arousal and unpleasant feelings [5]. Despite predominantly subjective reports of low arousal, boredom has been demonstrated to be physiologically a high state of autonomic arousal, i.e., increased electrodermal activity (EDA) and heart rate (HR; for a review, see [6]). Possible accompanying feelings include discomfort, resentment, anxiety, dissatisfaction, and frustration. These unpleasant feelings may be accompanied by an experienced lack of control, interest, goals, and/or motivation, disengagement, mental inactivity, and a desire for stimulation. Boredom results from the interaction of certain environmental factors (e.g., under- or overstimulation, monotony, a lack of challenge) and the individual tendency to feel bored under such circumstances [1,5,7]. This tendency is called “trait boredom” or “boredom proneness” and describes the individual propensity to experience states of boredom [8].

From a perspective of underlying mental processes, boredom occurs when people want, but are not able, to synchronize attention with either external (environmental) or internal (thoughts or feelings) information that is required to pursue an activity [7]. The unpleasant state of being bored is subsequently attributed to the environment, which is seen as boring or not offering opportunities to engage in [9]. The situation is perceived as lacking challenge and purpose [10], leading to a reorientation (pursuing alternative goals) or doing something different [6]. People usually seek to mitigate negative feelings while experiencing boredom by altering the perceived tedious activity, situation, or environment [11].

Phenomenologically, boredom is accompanied by negative emotions, difficulty concentrating, constraint, disrupted agency, low arousal, and a perceived slow passage of time leading to overestimations of the elapsed time [6,9]. The German word for boredom, *Langeweile*, meaning “long while”, refers to a perceived slow passage of a period. Accordingly, “…boredom is the experience of being disengaged and stuck in an endless dissatisfying present” [9] (p. 484). Supporting this perspective of a disturbed sense of time, Wangh [12] proposed that boredom is essentially experienced when the past, present, and future cannot be distinguished. Time appears as an endless present, causing a sense of impatience.

Boredom is associated with a variety of problematic behaviors and a reduced quality of life. For example, boredom and curiosity were found to be the most common causes of drug use [13,14] and risk-taking behavior in general [15]. Boredom is also associated with problematic smartphone use [16] and eating in obese and normal-weight persons [17]. The ability to cope with perceived boredom is reversely related to psychopathologies, such as depression and anxiety disorders [18]. In a professional context, boredom is highly related to job dissatisfaction [19], poor work performance [20], and task-unrelated thoughts, known as daydreaming [6,21]. In educational contexts, it was shown that boredom negatively affects learning [22] and also constitutes a motivational barrier [11]. It is, therefore, not surprising that proneness to boredom is moderately negatively related to life satisfaction [8]. A positive aspect is that boredom functionally serves as a signal to initiate changes that are more motivating than the current activity or situation [23]. It was also recently shown that boredom, inducing challenge-seeking behavior, may enhance creativity [24].

Farmer and Sundberg [8] developed the Boredom Proneness Scale (BPS) to capture the individual tendency to experience states of boredom. The BPS was found to be positively associated with negative affect (e.g., depression, anxiety, anger, hostility), unproductivity, perceived effort, loneliness, risky behaviors (e.g., aggressive driving, gambling, and hypersexuality), inattention, and impulsivity. The BPS is also negatively related to job performance and life satisfaction [8,20]. Eastwood et al. [25] found associations suggesting that more boredom-prone individuals are unaware of their emotions and rather externally oriented.

The 10-item subscale of the Sensation Seeking Scale (SSS) is the only general trait assessment of boredom that exists next to the BPS; all other trait boredom questionnaires are context-specific [26]. The “boredom susceptibility” subscale of the SSS is primarily related to externalizing issues (e.g., gambling, aggression) and “it appears to be related to sensation-seeking rather than boredom per se, and is thereby limited as a trait measure of boredom” [20] (p. 205). Additionally, due to the problematic reliability of the SSS and its exclusive focus on one major aspect of boredom, insufficient external stimulation [20], we opted for the BPS.

### 1.2. Consideration of Future Consequences

People frequently face a dilemma between short-term and long-term interests in their attitudes and behaviors, for example, for risk-taking [27], health behavior such as engaging in exercise or eating healthily [28], pro-environmental behavior and sustainability [29,30], aggressiveness [31,32], and delinquent behavior [33], as well as in financial investment decisions [34,35]. In general, in these kinds of temporal dilemmas, pursuing future goals (e.g., performing well in exams [36]) involves self-control: it is important not to give in to the alluring, but, in the long run, hindering, hedonistic activities in the present. In all these aforementioned topics, short-term gains conflict with undesired long-term consequences. One has to sacrifice pleasurable activities in the present to receive benefits in the long run. Studies have demonstrated that predictors for a higher consideration of long-term interests are reflected in stronger future orientation and a mindful present perspective. In contrast, hedonistic and impulsive present orientations are linked to making decisions that promote short-term interests [30,37].

Individuals differ in their present and future orientations when reaching behavioral decisions; this is considered to be a relatively stable, trait-like individual characteristic. Some individuals emphasize their immediate desires and tend to act to satisfy these needs. Some people focus on the outcomes of their present behaviors and consider alternative actions with regard to achieving more distant goals. Building upon the two-factor model of consideration of future consequences [38], Joireman et al. [28] developed a revised version of the Consideration of Future Consequences Scale (CFC-14), which specifically assesses how consideration of future outcomes affects decisions on present actions.

Corroborating the construct validity of the CFC-14, it was shown that individuals with high scores on the “future” subscale (CFC-F) were more engaged in exercise and healthy eating, i.e., more promotion-oriented [28], whereas problematic gamblers exhibited higher scores on the “immediate” subscale (CFC-I) [39,40]. CFC sum scores (irrespective of the subscales) were found to be positively associated with conscientiousness and delay of gratification, reflecting higher self-control [41]. The CFC is also negatively correlated with impulsivity [42]. For a more extensive review of findings regarding the CFC in domains like health, financial decision-making, work, environmental behaviors, and ethical decisions, see Joireman and King’s [43] recent publication. Notably, individual subjective time orientations are related to the subjective passage of shorter time intervals in the range of minutes [1,3], as well as questions about the subjective passage of time in everyday life [44].

One other inventory exists that captures temporal orientations, the Zimbardo Time Perspective Inventory (ZTPI) [37]. These questionnaires are not mutually exclusive, but typically show only low to moderate correlations [45,46,47] and have a rather distinct focus on the time perspective: the ZTPI reflects a broader and more normative view on the past, present, or future, whereas the CFC-14 more specifically and explicitly addresses how the time perspective shapes decisions and actions in the here and now. The shorter CFC-14 showed higher reliability than the ZTPI [45] and predicted health behaviors and intentions more effectively [47].

### 1.3. Self-Awareness

Interoception concerns physiological sensations permitting the brain to create a sense of the internal physical state [48]. This process is an important basis for several other human functions. For instance, theories of emotion posit that interoceptive processing contributes to emotional experiences by combining the information from the peripheral nervous system about the visceral and somatic changes with sensory information [49,50]. The insular cortex is the primary interoceptive area in the brain essential for monitoring of the current physical condition and emotional processing [51,52]. The insula also plays an important role in experiencing the passage of time, which derives from an accumulation of internal bodily signals over time [53]. Interoception is an important construct that serves as a factor in the experience of one’s own body, emotions, and time.

Garfinkel and Critchley [54] proposed three facets of interoception which have to be carefully distinguished from each other. These facets were concisely defined by Garfinkel et al. [55] to counteract the confusion caused by previously often interchangeably used terms, which also complicated comparisons among studies [56]. Interoceptive *accuracy* describes performance in objective tests of bodily sensations (e.g., how accurately the heartbeat is perceived in the heartbeat-perception test), whereas interoceptive *awareness* (IA) refers to the metacognitive evaluation of how confident an individual is when rating bodily sensations [55,56]. Interoceptive *sensibility* refers to the tendency to be self-focused on internal body states and is typically assessed with self-report measures, like questionnaires.

Longarzo et al. [48] originally developed the Self-Awareness Questionnaire (SAQ) in Italian. The SAQ is a scale assessing the self-evaluation of how often a person consciously experiences bodily sensations. SAQ scores were found to be associated with alexithymia, anxiety, and hypochondriasis, showing that these disturbances arise due to the misinterpretation and amplification of one’s bodily sensations [48,57]. Lower scores on a precursor questionnaire (on which the SAQ is based on) are further associated with damage in the insula [58], and IA, as measured with the SAQ, was shown to be correlated with insular connectivity [59]. These findings corroborate that the SAQ assesses an interoceptive component.

Teghil and coworkers [60] recently found that IA, as measured with the SAQ, predicts the timing accuracy (within the seconds range) in a condition lacking reliable, regular timing cues and that this relationship was mainly demonstrated by the subscale referring to visceral sensations. When the external environment does not provide a reliable time cue, human time perception seems to rely on bodily changes to indicate the passage of time. Timing accuracy then depends on the individual extent of interoceptive awareness for bodily (perhaps mainly visceral) processes. This finding supports the notion that human time perception is embodied and depends on IA or that IA at least contributes to time perception [61].

With the extensive 37-item Multidimensional Assessment of Interoceptive Awareness Version 2 (MAIA-2) [62], a conceptually related inventory exists. The eight subscales and the items, however, are not exclusively related to perceived interoceptive sensations. The MAIA-2 is a much more multifaceted questionnaire: it assesses how bodily sensations are treated cognitively (e.g., worry about, ignore, or distract from uncomfortable sensations; trusting one’s body) and includes the assessment of the ability to regulate distress by focusing attention to bodily sensations. It does not focus on specific sensations as does the SAQ, but asks for broader categories of, for example, body changes or discomfort. Therefore, the SAQ captures the construct of interoceptive awareness for diverse bodily sensations, whereas the MAIA-2 does not. Additionally, recent evidence presented in the above paragraph shows the importance of the SAQ in time perception research [60], which is our focus.

### 1.4. Metacognition of Time

People’s beliefs about how human time perception works and what factors influence it can affect their time estimations [63]. Sackett et al. [64] conducted a series of studies investigating the effects of metacognitive theories on time perception, i.e., naïve theories about how our sense of time works. The authors found that perceived time distortion influences the hedonistic evaluation of experiences—the felt passage of time serves as a metacognitive cue that is implicitly attributed to the enjoyment of the situation (“You’re having fun when time flies.”). Lamotte and coworkers [65] investigated how temporal judgments are affected by consciously considering the attention directed to time. The accuracy of time judgments exhibited fewer distortions the more aware the participants were of their focus of attention affecting time perception. Another study showed that the manipulation of knowledge about the effects of emotion (true, false, or no information) on the experience of time modulated the intensity of the impact of emotional stimuli on temporal judgments. The effect was either amplified with true information or attenuated after providing information contradicting actual empirical observations [66]. These studies show that adults are aware of temporal illusions that influence their time perception. The extent of their awareness of this factor, in turn, leads them to adjust their temporal judgments. Classic questionnaires on time perception do not merely assess the direct experience of time; time judgments are also a result of logical inferences made according to people’s metacognitive theories. That is why Lamotte et al. [63] developed the Metacognitive Questionnaire on Time (MQT) to assess individual consciousness of factors of time distortions that have been identified empirically.

## 2. Materials and Methods

### 2.1. Translation of the Questionnaires

The study aimed to validate German versions of the abovementioned questionnaires. A back-translation method was used to create the German versions to most accurately reproduce the content of the items in the original language of the questionnaires. The original versions were first translated independently by two native German researchers. These two slightly different versions were reviewed, and an adequate final version was agreed upon. This first German translation was then translated back into either English (CFC-14 and BPS), Italian (SAQ), or French (MQT) by two native speakers who were not familiar with the original version. The retranslated version was subsequently compared to the original, a revised second translation into German was constructed, and a consensus on a final version was reached, which was subsequently used in the present study (see Appendix A).

### 2.2. Procedure

The study was administered on an online survey platform (UniPark EFS Survey [67]). After being informed about the conditions of the study and the data processing, participants declared their written informed consent. Demographic data, including age, sex, native language, knowledge of German, and educational level, were recorded. The four questionnaires (BPS, CFC-14, SAQ, and MQT) were then assessed in random order. The participants were thanked and offered the opportunity to take part in a raffle for ten EUR 30 vouchers at a bookstore. The study was approved by the local Ethics Committee of the Institute for Frontier Areas of Psychology and Mental health (IGPP-2019-02).

### 2.3. Demographic Data and Time on Task

The participants’ ages and native languages were recorded, and participants were also instructed to rate their knowledge of German on a four-point scale (1 = fluent, 2 = advanced, 3 = good basic knowledge, 4 = beginner). Education was assessed on a five-point scale: 1 = no school-leaving certificate; 2 = *Hauptschule* (secondary school from grade 5 to 9); 3 = *Realschule* (secondary school from grade 5 to 10); 4 = *Abitur* (university-entrance diploma); 5 = university degree. Time on task, i.e., the time needed to complete the survey, was measured from opening the survey until reaching the last page with the raffle.

### 2.4. Participants

Subjects were recruited by advertisements on the local university website, emails to seminar participants at cooperating universities, and by word of mouth. Inclusion criteria were at least 18 years of age and fluency in the German language. In total, 314 participants completed all four online questionnaires (219 females (69.7%), 94 males (29.9%), 1 diverse (0.3%); aged μ = 31.07 years, SD = 12.43, range: 17 to 83). The final sample consisted of N = 273 subjects, since the data of 41 participants were excluded from the analyses (listed subsequently): nineteen of these excluded individuals showed unrealistic times on the task with less than 500 s, that is, less than 5 s spent on each of the 101 questionnaire items in total for reading and scoring. Twenty excluded participants interrupted the task at least once, making it impossible to determine the time on task. One participant whose data were dismissed from analyses spent an exaggerated time of 8172 s (2 h 16 min) on the questionnaires. Data of one further participant were removed because she reported being a beginner in learning German. Table 1 lists the demographic data of the remaining N = 273 (187 female (68.5%), 85 male (31.1%), and 1 diverse (0.3%) participants.

The distribution of the participants’ ages was right-skewed, with a majority of participants under 30 years of age (66.3%) and only 13.2% (36 participants) over 50. Nevertheless, the wide range from 17 to 83 indicates that people from a large age spectrum participated in our study.

Knowledge of German was high; 261 participants called German their native language. Among the twelve remaining participants with a different mother tongue, twelve reported speaking German fluently, and one admitted having “only” advanced knowledge of German.

The educational levels were high on average, with 156 (57.14%) university graduates, 109 (39.93%) with Abitur (German high-school graduation), and only eight individuals with a Realschulabschluss (a school-leaving certificate usually taken after the fifth year of secondary school in Germany).

On average, it took 16 min and 16 sec to complete the questionnaires, but there was high inter-individual variance in duration. There was a positive correlation between age and the time it took participants to complete the four questionnaires, *r* = 0.22, *p* < 0.001, indicating that the older the participants were, the more time they needed to fill in the survey.

### 2.5. Questionnaires

#### 2.5.1. Boredom Proneness Scale (BPS)

The Boredom Proneness Scale (BPS) [8] assesses the individual tendency to experience situations as boring or monotonous. The original English version consists of 28 items subdivided into two to five subscales, depending on the factor-analytic study. Vodanovich and Kass [68] proposed the following five factors: lack of external stimulation, lack of internal stimulation, affective responses, constraint (or waiting behavior), and perception of time. The items are rated on a 7-point Likert scale. We inverted the poles of this original scale so that a higher level corresponds to a higher number (1 = highly disagree; 7 = highly agree) as in the other administered questionnaires to avoid accidental wrong answers. A sum score was calculated. The reliability of the original English BPS was satisfactory with an internal consistency of α = 0.79 to 0.91 [20], and subscale reliabilities were low to adequate with α = 0.59 to 0.73. Test–retest reliability over one week was found to be high in an English sample (*r* = 0.83), and construct validity was verified in various tests [8]. It should be mentioned that there are both a 12-item and 8-item short form available (BPS-SF) [69,70]. The original aim of developing the questionnaire was to capture “…one’s connectedness with one’s environment on many situational dimensions, as well as the ability to access adaptive resources and realize competencies” [8] (p. 10).

#### 2.5.2. Consideration of Future Consequences Scale–14 (CFC-14)

The Consideration of Future Consequences Scale was originally developed by Strathman et al. [41] to assess the extent to which potential distant outcomes are considered and how this affects decisions on current behavior. This original 12-item inventory was previously generally treated as a unidimensional scale [41]. Joireman and colleagues [38] distinguished two dimensions: one subscale relating to concern with immediate consequences (CFC-I) and the other assessing concern about (temporally more distant) future consequences (CFC-F). Joireman et al. [28] introduced two additional items into this subscale to improve the suboptimal reliability of the original 5-item CFC-F subscale. The original 5-point scale was extended to a 7-point scale to increase variance. The two-factor structure of this updated English 14-item version was supported using exploratory and confirmatory factor analyses and showed high internal consistency values of α = 0.80 (CFC-F) to α = 0.84 (CFC-I) [28].

The CFC-I and CFC-F subscale scores are sum scores of the respective allocated items. An overall CFC sum score results from adding the scores on the CFC-F items and the inverted CFC-I item scores as originally proposed by Strathman et al. [41] to have a composite score for future orientation. However, the underlying assumption thereby is that by strongly agreeing with a CFC-I item, one would at the same time disagree with a CFC-F item. Joireman et al. [28] deviated from the unidimensional concept: The CFC-I score indicating present orientation and the pursuit of smaller, immediate rewards and the CFC-F score reflecting future orientation and seeking larger, but often less certain gratification, were considered to be more reliable as separate dimensions.

#### 2.5.3. Self-Awareness Questionnaire (SAQ)

The Self-Awareness Questionnaire (SAQ) is a 28-item self-report inventory assessing the frequency of common bodily states and sensations [48]. The authors aimed to capture interoceptive awareness. In contrast to Garfinkel and colleagues’ [55] definitions, interoceptive awareness here describes cognitively interpreted interoceptive sensations, as well as the ability to express bodily feelings. The items are rated on a 5-point Likert scale ranging from 0 = never to 4 = always. The total score ranges from 0 to 112, and higher scores reflect a higher degree of interoceptive awareness.

The questionnaire has shown a good internal consistency of Cronbach’s α = 0.88 in the original Italian version. It consisted of two factors, the first related mainly to visceral sensations, and the second factor referring mainly to somatosensory sensations (F1: α = 0.85; F2: α = 0.81). In the English validation, the questionnaire demonstrated excellent internal consistency of α = 0.90. Allocations of the items to the two factors in the factor analysis of the English questionnaire differed from the Italian version [57].

#### 2.5.4. Metacognitive Questionnaire on Time (MQT)

The Metacognitive Questionnaire on Time (MQT) [63] comprises 24 items evaluated on a 5-point Likert scale (1 = totally agree; 5 = totally disagree). It was designed to assess “interindividual variations in the awareness of factors affecting the experience of the passage of time” in oneself and others [63] (p. 339). The questionnaire is subdivided into a “self” scale and an “others” scale. Each of these scales consists of two factors, attention (8 items) and emotion (4 items), which reflect time-perception factors that have been identified in empirical studies. The reliability of the factors in each scale was found to be adequate to satisfactory (Cronbach’s α = 0.68 to 0.81) in the original French version. A second study seeking to validate the original French MQT again showed low reliability for the “self” scale (α = 0.66; attention: α = 0.67; emotion: α = 0.50). However, in this second study, the “others” scale suggested that the knowledge or estimations of others’ temporal distortions are not reliable because confirmatory factor analyses were not able to fit the data well [63]. A possible interpretation could be that people find it difficult to estimate how time perception in others works. Construct validity was also verified with several tests on convergent and discriminant validity. In the German version, the scales were reversed (1 = totally disagree; 5 = totally agree) to produce a consistent scale orientation for all four administered questionnaires and prevent confusion.

### 2.6. Statistical Analyses

#### 2.6.1. Exploratory Factor Analyses

For validation of the questionnaires, the factorial structure was first examined using unweighted least squares (ULS) as the extraction method in the exploratory factor analysis (EFA). ULS does not assume a multivariate normal distribution. Factors are latent dimensions accounting for the greatest possible variance in the original manifest variables (questionnaire items). The factors are integrated into a model designed to be parsimonious. We used the Pearson correlation matrix following the recommendations for EFAs in validation analyses by Izquierdo and coworkers [71]. Prior to the analysis, the suitability of the data for the factorial analysis was checked using the Kaiser–Meyer–Olkin measure (KMO) and Bartlett’s test for sphericity to probe for acceptably large correlations [72]. A parallel analysis (PA) was conducted to determine the number of factors to be retained [73,74]. The PA is considered an accurate tool to detect significant components and to identify the corresponding item loadings, whereas the eigenvalue > 1 criterion has been shown to lead to an over-extraction of factors [75,76]. Therefore, in the PA, the eigenvalues from the raw data of the questionnaires were compared to 5000 simulated random datasets of non-correlated, normally distributed variables. These random data coincide with the raw data in that they comprise an equal number of variables and the same sample size. Significant factors in the PA explain more variance in the actual data than 95% of the random datasets [77]. Items were removed from the German versions of the questionnaire when their communality was below h^2^ < 0.30 and the factor loadings were smaller than λ = 0.40 on all factors [78]. That is, when both exclusion criteria were met, items were removed. Oblique rotation (direct oblimin) with Kaiser normalization was conducted because this rotation method does not restrict factors to be uncorrelated, but results in a factor solution similar to an orthogonal one [79]. Items with factor loadings of λ ≥ 0.40 were then allocated to the respective factor. For factor loadings λ ≥ 0.40 on multiple factors (cross-loadings), we thoroughly examined the content of the respective items and subsumed them either under both subscales (where the content reflected an interaction of both and was therefore interpretable) or to one of the subscales when the item’s content indicated this unique allocation.

#### 2.6.2. Intercorrelations and Internal Consistencies

The internal consistency of the scale, an indicator of reliability, was calculated using Cronbach’s alpha with values varying between −∞ and 1. Values of α > 0.70 are interpreted as acceptable [80,81]. Due to problems with Cronbach’s alpha having unrealistic assumptions that lead to an underestimation of internal consistencies, we included further measures of reliability, namely omega total (ω_total_), Revelle’s ω_total_, and the greatest lower bound (GLB) [82]. It was also controlled whether the sorting of items influenced Cronbach’s α. The result is not described when the deletion of items would have reduced or not affected Cronbach’s α. Pearson correlations were calculated between the questionnaires to elucidate the associations between the constructs.

All statistical analyses were performed using IBM SPSS Statistics version 20 except for the additional internal consistency indices, which were calculated using the freeware statistics program R (https://www.r-project.org).

## 3. Results

### 3.1. Boredom Proneness Scale (BPS)

Two participants were outliers (Boxplot criterion), but these were not unrealistic raw scores. The two participants were, therefore, included in the analyses representing very boredom-prone individuals.

The measure of sampling adequacy (MSA; Kaiser–Meyer–Olkin, KMO) was very good; KMO = 0.84. The Bartlett test for sphericity showed significance with χ^2^ = 2240.15, df = 378, *p* < 0.001. These characteristics indicated that the data were suitable for the EFA.

#### 3.1.1. Three-Factorial EFA of the BPS

For the 28-item BPS, the initial parallel analysis identified three factors to be extracted. Communalities h^2^ < 0.30 were observed for thirteen items (2, 3, 5, 6, 7, 9, 17, 18, 22, 23, 24, 27, 28). Five of these items had factor loadings of less than λ = 0.40 (items 3, 6, 7, 22, 27). These items were therefore removed. Items 2, 18, and 28 were subsequently additionally removed for the same reasons. The PA on the reduced 20-item data still suggested three factors to be retrieved (see scree plot in Figure 1). The three-factor solution suggested by the PA explains 46.79% of the variance. An overview of the final EFA can be seen in Table 2.

#### 3.1.2. Internal Consistency of the German BPS-20

According to this three-factor solution, the first factor subsumed eleven items (items 1, 5, 9, 10, 11, 12, 19, 20, 21, 25, 26) with an internal consistency of α = 0.84 (ω_total_ = 0.84, 95%CI [0.81, 0.87]; Revelle’s ω_total_ = 0.87; GLB = 0.90). The second factor contained ten items (1, 4, 8, 10, 11, 13, 14, 16, 23, 24) with α = 0.83 (ω_total_ = 0.83, 95%CI [0.79, 0.86]; Revelle’s ω_total_ = 0.87; GLB = 0.92). Two items (15, 17) loaded high on factor three; Cronbach’s α = 0.64 (ω_total_ = 0.65, 95%CI [0.55, 0.75]). This value is acceptable since this factor replicates the subscale “constraint” of Vodanovich and Kass’ [68] factorial analysis. The first factor (“external”) refers to a lack of external stimulation and excitement. The second factor (“internal”) concerns the propensity of internal stimulation or the ability to entertain oneself. Discriminatory power was suboptimal with λ > 0.40 on more than one factor for items 1, 4, 10, 11, and 14. We allocated items 1, 10, and 11 to both factors referring to the content of the items which describe conjunctions of external and internal (under-)stimulation (see, for example, item 11: “Ich finde Gefallen an den meisten Dingen, die ich tue”/English original: “I get a kick out of most things I do”). The remaining two items (4, 14) were allocated to the second factor because they clearly described a lack of internal stimulation (“sit around doing nothing”/“not knowing what to do”). The internal consistency of the whole BPS-20 questionnaire with 20 items remaining is good with Cronbach’s α = 0.86 (ω_total_ = 0.86, 95%CI [0.83, 0.89]; Revelle’s ω_total_ = 0.89; GLB = 0.92). The BPS-20 sum score and the respective scores on the three subscales derived from the EFA can be seen in Table 3.

### 3.2. Consideration of Future Consequences Scale-14 (CFC-14)

KMO = 0.84 is very good, and the Bartlett test is significant with χ^2^ = 1094.46, df = 91, *p* < 0.001. The initial PA on the CFC-14 suggested two factors to be retained.

#### 3.2.1. Two-Factorial EFA of the CFC Scale

Low communalities of h^2^ < 0.30 were observed after extraction for items 5 (h^2^ = 0.13), 6 (h^2^ = 0.15), 8 (h^2^ = 0.25), 12 (h^2^ = 0.15), and 13 (h^2^ = 0.27). There were low factor loadings of λ < 0.40 for items 5, 6, 8, and 12, which were therefore removed, leaving five items for each subscale. The two-factor solution recommended by the PA (see Figure 2) was still unsatisfactory; item allocations to factors were unambiguous for only half of the items (double loadings for items 3, 4, 11, 13, and 14) and in spite of the inverse relation of CFC-I and CFC-F, the factor loadings were either positive or negative for both factors. For an overview of this two-factorial EFA solution, see Appendix A.

#### 3.2.2. Single-Factor EFA of the CFC Scale

Assuming a single factor as proposed earlier [41] and in spite of the PA recommending extraction of two factors (see Figure 2), low communalities of h^2^ < 0.30 were present for items 1 (h^2^ = 0.26), 5 (h^2^ = 0.13), 6 (h^2^ = 0.16), 7, 8 (both h^2^ = 0.25), 9, 10 (both h^2^ = 0.26), 12 (h^2^ = 0.10), and 13 (h^2^ = 0.28). After again removing items 5, 6, 8, and 12 because of low factor loadings of λ < 0.40, 39.12% of the variance can be accounted for by this factor. Negative factor loadings exactly replicate the CFC-F subscale, whereas CFC-I items exhibit highly positive loadings (see Table 4). This supports the (at least) statistical legitimacy of calculating the sum score of CFC-F and inverted CFC-I items.

#### 3.2.3. Correlation of the CFC Subscales

As expected, the two reduced theoretical subscales CFC-I (without items 5 and 12) and CFC-F (without items 6 and 8) were highly negatively associated with a correlation of *r =* −0.582, *p* < 0.001 for the reduced CFC-I and CFC-F scales. This inverse relation corroborates the structure of two subscales with opposing orientations (also reflected in the inverse polarity of the factor loadings of the unidimensional CFC-10). We encourage the use of the CFC-F and CFC-I subscores as meaningful indicators next to the statistically retrieved unidimensional CFC sum score.

#### 3.2.4. Internal Consistency of the German CFC-10 Scale

Cronbach’s alpha indicating internal consistency was acceptable with α = 0.76 for the 5-item CFC-I immediate subscale (ω_total_ = 0.76, 95%CI [0.72, 0.81]; Revelle’s ω_total_ = 0.83; GLB = 0.81), and α = 0.71 for the shortened CFC-F future subscale assessed consideration of more remote consequences (ω_total_ = 0.72, 95%CI [0.66, 0.79]; Revelle’s ω_total_ = 0.82; GLB = 0.81). The internal consistency was good for the whole CFC-10 scale with Cronbach’s α = 0.83 (ω_total_ = 0.82, 95%CI [0.79, 0.86]; Revelle’s ω_total_ = 0.87; GLB = 0.89) for the single-factor scale and, importantly, exceeded the values for the proposed subscales.

Descriptive statistics are demonstrated in Table 5, showing that the mean sum score of the CFC-future subscale was higher than the mean sum score of the CFC-immediate subscale.

### 3.3. Self-Awareness Questionnaire (SAQ)

The KMO was very good (KMO = 0.86) for the 28-item SAQ proposed by Longarzo et al. [48], and the Bartlett test was significant with χ^2^ = 1808.52, df = 378, *p* < 0.001. The parallel analysis suggested two factors as also observed in the English and Italian samples [48,53].

#### 3.3.1. Two-Factorial EFA of the SAQ

Communalities h^2^ < 0.30 were observed for eighteen items (1, 2, 3, 4, 5, 6, 7, 10, 11, 12, 13, 19, 22, 23, 24, 25, 27, 28). Nine of these items (1, 3, 5, 11, 12, 13, 22, 24, 27) also exhibited factor loadings < 0.40 on both factors; these were deleted. Subsequently, items 2 and 19 were removed for the same reasons of both low communalities and factor loadings. The overview of the following EFA is shown in Appendix A.

The two retained factors accounted for 38.80% of the variance. The first factor included all items, whereas six items could also be allocated to factor 2. Internal consistency was good with α = 0.84 for the first factor and α = 0.70 for the second factor. However, there was a huge number of items lacking discriminatory power (λ > 0.40 on both factors for eight items: 4, 8, 9, 13, 20, 21, 22, 25). The varying contents of the items subordinated to the respective factors did also not integrate specific subcomponents of the construct of interoceptive awareness. This supports a unidimensional structure of the SAQ in German. This is also in line with the observation in the Italian and English versions [48,53], where items were differently allocated to the factors and their labeling as “mainly” relating to visceral vs. somatosensory feelings was arbitrary and merely statistical rather than referring to specific content clusters. Even in the two Italian samples, the item allocations could only be partially replicated (18/28; 64%).

#### 3.3.2. Single-Factor EFA and Internal Consistency of the German SAQ-17

Assuming a single factor instead, communalities h^2^ > 0.30 were only observed for five items—8, 9, 18, 20, 26. Factor loadings < 0.40 were present for items 2, 3, 5, 6, 11, 13, 15, 19, 22, 24, and 27, which were therefore removed resulting in a SAQ-17 scale. Figure 3 shows that for this shortened scale, a single-factor solution was appropriate. The single factor accounted for 31.47% of the variance (see Table 6 for the EFA). The internal consistency value for the whole SAQ scale was high with Cronbach’s α = 0.86 (ω_total_ = 0.86, 95%CI [0.82, 0.89]; Revelle’s ω_total_ = 0.89; GLB = 0.91) and thus, even higher than the subscale internal consistency indices, which corroborates the unidimensional solution. The sample of the present study showed a μ = 29.49 (SD = 6.73; range: 18–61) on the German SAQ-17.

### 3.4. Metacognitive Questionnaire on Time (MQT)

The MQT consists of two scales which are conceptually different, one investigating metacognitive beliefs about one’s own time perception, and the other referring to how others might think about their time perceptions. Therefore, two separate EFAs were conducted on these scales.

#### 3.4.1. MQT-Others Subscale

##### EFA on the MQT-Others Subscale

The MSA was appropriate with a value of KMO = 0.69; the Bartlett test for sphericity was significant; χ^2^ = 338.48, df = 66, *p* < 0.001. The PA proposed the extraction of two factors (see Figure 4).

Communalities h^2^ < 0.30 were observed for all items except items 1, 18, and 23. There were factor loadings > 0.40 for six items: 1, 9, 12, 17, 18, and 23. Item 9 was subsequently deleted for the reason of low communality and factor loadings. The EFA on the remaining five items suggested that factor 1 consists of items 12, 17, and 18 (all originally “attention” items); items 1 and 23 were allocated to the second factor (one “attention”, one “emotion” item; Appendix A). The extraction of a single factor additionally proposed the deletion of item 12. This factor contained three items originally constituting the “attention” factor and one “emotion” item (see Table 7). Internal consistency analysis showed a poor Cronbach’s alpha of α = 0.61. This EFA solution was unsatisfactory, recommending the deletion of the majority of the items and still having poor communalities and reliability. Acknowledging the content of the items which derive from scientific evidence (factors that affect human time perception), we decided to focus on maximization of internal consistency instead and discuss these results.

##### Internal Consistency and Final German Solution of the MQT-Others Subscale

Cronbach’s alpha of the original 12-item “others” scale was α = 0.54 (ω_total_ = 0.52, 95%CI [0.40, 0.64]; Revelle’s ω_total_ = 0.67; GLB = 0.72). When items 3 and 10 (one belonging to the “attention” and one to the “emotion” subscale, respectively) are omitted (Appendix A), this value can be raised to a maximum of α = 0.62 (ω_total_ = 0.60, 95%CI [0.50, 0.70]; Revelle’s ω_total_ = 0.71; GLB = 0.74). The internal consistency was still not entirely satisfactory but was similar to the reliabilities found in the original version [63]. The descriptive statistics of the shortened MQT-“others” subscale and its respective factors “attention” and “emotion” (with item allocations as in the theoretically founded original French version [63]) are shown in Table 8.

#### 3.4.2. MQT-Self Subscale

##### EFA on the MQT-Self Subscale

The KMO = 0.71 and the significant Bartlett test (χ^2^ = 432.64, df = 66, *p* < 0.001) indicated suitability of the data for factorial analysis. Retention of two factors was suggested by the PA (see Figure 5).

Communalities were lower than h^2^ = 0.30 except for items 4, 14, 15, and 20. For half of the items, factor loadings > 0.40 were present: items 4, 5, 14, 15, 20, 21, and 22. Items 21 and 22 were removed subsequently for low communalities and factor loadings. Factor 1 subsumed all items except for item 5, whereas factor 2 contained items 4, 5, and 20. Items 4 and 20 had double loadings with λ > 0.40 on both proposed factors (Appendix A). A single factor fitted the data better than a two-factorial solution after removing these items. The five-item scale proposed by the EFA contained three “attention” and two “emotion” items; Cronbach’s α = 0.65 (Table 9). As for the “others” subscale, we subsequently focused on the maximization of internal consistency.

##### Internal Consistency and Final German Solution of the MQT-Self Subscale

The complete original 12-item “self” subscale shows a poor internal consistency of α = 0.52 (ω_total_ = 0.55, 95%CI [0.47, 0.62]; Revelle’s ω_total_ = 0.72; GLB = 0.75). When item 16 (“attention”) and 19 (“emotion”) were deleted, the internal consistency rose considerably to α = 0.65 (ω_total_ = 0.65, 95%CI [0.59, 0.71]; Revelle’s ω_total_ = 0.73; GLB = 0.76). Further simulations showed that by additionally deleting items 5, 6, and 24, the internal consistency increases only slightly and maximally to α = 0.68 after sorting out nearly half (5/12) of the items of the “self” subscale. The internal consistency was still not entirely satisfactory, but was similar to the previous reliability analyses [63]. The descriptive statistics of the 10-item MQT-“self” subscale (items 16 and 19 omitted) and its respective factors “attention” and “emotion” (with item allocations as in the original version [63]) are shown in Table 8.

#### 3.4.3. Internal Consistency of the Whole MQT and Intercorrelation of Its Subscales

The internal consistency for the whole 20-item scale was acceptable; α = 0.78 (ω_total_ = 0.76, 95%CI [0.72, 0.81]; Revelle’s ω_total_ = 0.82; GLB = 0.87). The (shortened 10-item) MQT “self” subscale was associated with the (10-item) “others” subscale with *r* = 0.645, *p* < 0.001, reflecting a high overlap of factors which determine one’s own time perception and the sense of time in other human beings in their metacognitive beliefs.

### 3.5. Intercorrelations among the Questionnaires

CFC-I correlated with boredom proneness (BPS-20), *r* = 0.233, *p* < 0.001, whereas CFC-F was negatively associated with boredom proneness (BPS-20), *r* = −0.175, *p* = 0.004. The CFC-10 sum score also correlated with boredom proneness (CFC-10–BPS-20: *r* = −0.226, *p* < 0.001).

The greater their self-awareness as measured with the SAQ, the more boredom prone the participants were (SAQ-17–BPS-20: *r* = 0.330, *p* < 0.001).

The shortened 10-item “self” subscale was weakly associated with the SAQ17 (*r* = 0.160; *p* = 0.008).

Considering the inflation of alpha errors for accumulated testing, the false-discovery rate (FDR) [83] was applied as a correction method for multiple testing. With twenty-one correlations (between BPS-20, CFC-10, CFC-I, CFC-F, SAQ-17, MQT-“self”, and MQT-“others”, respectively), the reported correlations all survived the FDR correction.

## 4. Discussion

### 4.1. Summary of Validations and Theoretical Reflection of the Solutions

In the present study, we validated four questionnaires which had been translated into German, namely a scale assessing boredom proneness as a trait (BPS), an inventory gauging the consideration of future consequences in general decision-making (CFC-14), a questionnaire aiming to capture interoceptive awareness (SAQ), and a scale on metacognitive theories about human time perception (MQT).

Concerning the BPS, the exploratory factor analysis (EFA) suggested a three-factor structure. The first factor integrates items assessing lack of external stimulation, whereas the second factor gauges lack of internal stimulation. The third factor robustly replicates the “constraint” subscale identically found in the Kass and Vodanovich factor analytic study referring to waiting situations [68]. The factors find their theoretical basis in the notion that boredom is an interaction of environmental factors (e.g., monotony, a lack of challenge) and the individual inability to distract or entertain oneself in such situations [1,5,7]. The constraint factor with its strong focus on the (in)ability to wait relates to findings associating boredom and an altered sense of time, i.e., an overestimation of these waiting situations [6,9,12]. However, this “constraint” factor contains only two items, which is not very common for a factor. Preferably, it would consist of four or more items [71] “to provide minimum coverage of the construct’s theoretical domain” [84] (p. 676). Yet, for very narrowly defined constructs (as might be reflected here in the (in-)ability to wait), there is evidence suggesting that even single-item measures may be sufficient [85,86]. This factor may thus also be identified with only two items, although this is generally not recommended [87,88]. A factor consisting of two items should be interpreted with caution but may be considered reliable when these are highly correlated (here: Spearman’s ρ = 0.45, *p* < 0.001), but fairly uncorrelated with other variables (range: ρ = 0.002 to ρ = 0.246; μ = 0.093) of other factors. Nevertheless, it is more important than regarding the merely statistical solution, to retain a factor if it can be interpreted well by considering its content [89]: the items should address the same issue, revolving around the construct behind the factor (here: waiting situations) by asking a single question phrased in different ways. This is the case for this two-item factor “constraint”, which focuses on an individual’s ability to wait patiently (vs. not). More impulsive individuals as well as people with lower self-regulation capacities feel more negatively (i.e., bored) when having to wait and they feel a slower passage of time [1,3]. In a representative study by the German Society for Consumer Research (Gesellschaft für Konsumforschung, GfK) in 2016, it was found that the situation which annoys Germans most (55.6% of participants responded to this item) is “avoidable long waiting times”. Summing up, the content supports the three-factor solution found in the EFA, but the “constraint” factor with only two items assessing the individual’s (in-)ability to wait patiently should be interpreted cautiously and should be extended by at least two items in subsequent validations. Eventually, after deleting various items in the German version for reasons of low communality and factor loadings, the three-factor solution explains 49.45% of the variance of the resulting BPS-20.

The EFA of the German CFC-14 initially indicated a two-factorial structure. Some factor loadings were low, but positive on the designated factor (CFC-I; CFC-F) and highly negative on the other factor. A single-factor model as originally proposed [41], and by removing four items, fitted the data better. The CFC-F items exhibited high negative-factor loadings, whereas highly positively loading items replicate the CFC-I subscale, which reflects the inverse relationship found here (*r* = −0.58) and in previous studies [28,38]. This finding justifies calculating a CFC sum score with inverted CFC-I items as an index for consideration of future consequences. This general factor accounted for 39.12% of the variance. This inverse relation, the theoretical background, and item content justify also calculating the scores of the two subscales with opposing orientation for comparison with the respective scores in other samples (in different languages). The internal consistency was lower for the two subscales (CFC-I: α = 0.76; CFC-F: α = 0.71) than for the entire scale (α = 0.82). A single factor instead of two factors seems more appropriate for the CFC according to our statistical analyses, but the CFC-I and CFC-F subscores still appear to be meaningful indices.

The initial PA supported a two-factorial structure for the SAQ. However, a unidimensional structure seems more appropriate considering the arbitrary allocation of items to the factors and the low discriminatory power of the items allocated to the second factor. The PA on the shortened scale confirmed this solution. Theoretically, interoception is thought to reflect the awareness for signals from the peripheral nervous system about visceral and somatic changes. Neuroanatomical pathways differently process visceral and somatic information [51], and the insular cortex serves as a hub region for integrating the physiological inner signals and disseminating them into other brain areas for cognitive, emotional and motivational processing [90,91]. The present results suggest that individual interoceptive awareness might be more general for bodily sensations irrespective of their visceral or somatosensory origin. Corroborating the unidimensional solution, the internal consistency value for the whole reduced SAQ-17 scale was excellent with Cronbach’s α = 0.86 and even higher than the subscale alpha indices.

Analyzing the conceptually distinguishable subscales of the MQT separately in two PAs revealed a two-factorial structure for both. The EFA failed to replicate the proposed “emotion” and “attention” factors for both subscales and recommended deleting a majority of the items. Conspicuously, three out of the four (“others”) and five (“self”) items were matched: their content was exactly the same, one time phrased for the assessment of metacognitive beliefs about the others’ sense of time, one time for their own time perception (items 1–14; 4–23; 15–18). These might, therefore, reflect items that worked particularly well for the German sample in terms of interpretation and rating. All these items refer to the saying “time flies when you’re having fun” [64,92]. Considering that, it seems plausible that this factor on time perception is conceived as highly certain by the participants, whereas they might be more insecure about the other items and the impact of the respective factors on the sense of time. This might account for why a majority of items had low factor loadings and were therefore deleted. Osborne and Fitzpatrick pointed out the difficulty of replicating EFA results; even under optimal conditions [93], item deletion is very frequent [89]. However, irrespective of the unsatisfactory present statistical solution in the EFA, the MQT is a questionnaire developed on a conceptual and theoretical basis of scientific evidence on factors that influence the sense of time [63]. “Emotional arousal” and “attention directed to time” are the two important factors that have been identified as crucial to how the passage of time is perceived [94]. Focusing on internal consistency maximization instead, we removed two items of each subscale (one “attention”, one “emotion” item), resulting in a shortened version with ten items each. The internal consistencies of the “self” and “others” scales were still not entirely satisfactory. The reliability indices of the original French evaluation were similarly low [63]. The shortened 20-item MQT had a satisfactory internal consistency of α = 0.78. The 10-item subscales were highly correlated (*r* = 0.65), suggesting very similar metacognitive beliefs about one’s own and others’ time perceptions. When forming a notion of others’ perspectives on time, people probably resort to their own experience and generalize it to others. This phenomenon may be reflected in the well-known “false consensus effect”, referring to an overestimation of the extent to which one’s own opinions, habits, preferences, and values are shared by others [95]. Additionally, it was reported by several participants that this scale was confusing and that it was hard to separate between the evaluations of their own and others’ time perception. This might be also reflected in the high correlation between the two subscales.

### 4.2. Discussion of the Deletion of Items

There were many items deleted due to low communalities and factor loadings. This is particularly the case for the BPS (eight deleted items) and the SAQ (eleven items removed). Several reasons are imaginable for causing the shortening of these scales. Possibly, there may be differences in interpretation due to language or cultural differences compared to the original questionnaires. It might also be that these modifications are related to sample properties. We had a sample which was not completely representative (see limitations section: high proportion of females, comparably high education, and the age distribution) but which also differs to the samples consisting exclusively (BPS [8,68], CFC [28], MQT [63]) or mostly (SAQ [48]) of a young undergraduate student population. From a statistical view, an EFA is a data-driven procedure which is based on exploring the relationships between items (in terms of correlations or covariances) to uncover patterns to discover latent variables. Replicating EFA results even under optimal conditions is therefore very difficult and rare [93]. Worthington and Whittaker [89] state that “on rare occasions, a researcher may retain all the initial items submitted to EFA. Item deletion is a very common and expected part of the process” (p. 822). We deleted items that failed to contribute substantially to the factor solutions (in terms of communalities and factor loadings). It remains speculative why, at least for the BPS and the SAQ, there was such a high number of items to be deleted according to the applied criteria. Nevertheless, it is also a quality criterion of questionnaires to be shorter as long as they still exhaustively assess the same construct and do not neglect aspects of it (parsimony). This is discussed in the following.

The final German BPS-20 did not include eight items which in essence reflected “external” and “internal stimulation”. Strikingly, all items including “always” statements were deleted. Potentially, participants might not have answered with an adequate inter-individual variation as this answer category was too general (original items 3, 7, 27). Items 2 and 6 may trigger social desirability, referring to feeling bored when watching private photos of others or frequently engaging in work-unrelated things at work. Other removed items were complex to estimate reliably because they referred either to an event long ago (item 28), waking-up situations (item 18), or possible evaluations by others (item 22). For these reasons, it might be that these items are at least partly confounded and for this reason, did not contribute sufficiently to the factors of boredom proneness. Considering the content of these items, it does not seem probable that an essential aspect of boredom proneness was neglected by deleting these items. This interpretation is corroborated by the PA on the 28-item BPS, which likewise recommended retention of three factors, which is still the case after deletion of these eight items.

Four items were removed in the shortened CFC-10. These were equally distributed among the CFC-I and the CFC-F factors. It is not likely that an essential aspect pertaining to the consideration of immediate or future consequences was eliminated. Additionally, the EFA recommends a unidimensional CFC.

Similarly, for the SAQ, the content of the deleted items did not show an aspect that was not included in the retained items. No particular pattern of interoceptive awareness of certain body parts or feelings was detectable. The SAQ can be conceived as a unidimensional scale; the deleted items should not have restricted the validity of the shortened scale.

The EFA on the MQT recommended the deletion of a majority of the items. It was reported by several participants that the scale was confusing because of the metacognitive items requiring a rather high level of introspection. As for the removal of items, we therefore decided not to consider the EFA but to focus on internal consistency maximization. However, reliability was still not entirely satisfactory. Considering the EFA results, the validity and explanatory power of this scale should be further evaluated. Nevertheless, the content reflects factors on human time perception retrieved from conceptual and empirical evidence.

Summing up, the factor analyses only partially confirmed the factorial structure found for the questionnaires tested in previous languages and samples and were unable to fully replicate the item allocations. Nevertheless, the statistical solutions (and the content of the items subsumed to the respective factors) presented here also make sense considering the theoretical understanding of the constructs. The factorial structures found for the German versions of the questionnaires in the present investigation should be analyzed with confirmatory factor analyses with representative samples in future studies.

### 4.3. Discussion of the Intercorrelations

#### 4.3.1. Correlation of Consideration of Future Consequences (CFC) with Boredom Proneness (BPS)

We found the consideration of future consequences moderately negatively associated with boredom proneness. The conversely related subscales for considering more future vs. immediate consequences related differently to boredom proneness (negative correlation for CFC-F and positive for CFC-I), confirming the inverse relation of the subscales also in its relation to external constructs. More future-oriented persons also show less boredom proneness. The finding that highly boredom-prone individuals are more likely to engage in impulsive and higher risk-taking behaviors has previously been observed [8,96,97,98]. Watt and Vodanovich [98] found a highly positive correlation of 0.56 between boredom proneness and impulsivity. Boden [99] explains that the experience of boredom usually entails maladaptive, sensation-seeking behavior and searching for rapid stimulation and reward.

Frequently feeling bored as the experience of being unsatisfactorily stuck in the moment [9] might induce a general orientation to the present reflected in higher CFC-I and lower CFC-F and CFC sum scores. As time was perceived as passing slower in more boredom-prone individuals [92,100], this could be the basis of an extended present. The present orientation was likewise shown to be associated with a perceived slower passage of time, whereas a future orientation results in a perceived faster passage of time [3]. Confirming the present findings, relations between the BPS and impulsivity were already shown, while impulsivity per definition describes a pronounced present orientation with a strong urge to act immediately and with less ability to delay gratification [101].

More future-oriented persons may be also less prone to experiencing boredom since they make more plans and thereby always have something to do; they find themselves less often in situations they did not choose. There is evidence that individuals with higher CFC scores are more promotion-oriented, pursue positive gains, and see more purpose in their actions [28,102]. More present-oriented persons, on the contrary, may experience boredom when the environment itself does not yield much stimulation.

#### 4.3.2. Correlation of Interoceptive Awareness (SAQ) with Boredom Proneness (BPS)

There was a moderate to high correlation between interoceptive awareness and boredom proneness. This strong association can be accounted for from a time-perception perspective. There is accumulating evidence that time perception is embodied. Human beings have no dedicated sensory organ for perceiving the flow of time; we have to rely on external and internal cues to sense the passage of time. However, there is still a sense of time even under sensory deprivation, that is, exclusion of external stimulation [57]. Experimental studies have confirmed that time is experienced by directing awareness to our bodily status. For example, the more accurately participants were able to count their heartbeats, the more accurately they estimated time in a subsequent duration-reproduction task [103]. An fMRI study showed that activity in the insula (the primary interoceptive cortex) accumulated during the estimation of stimuli durations and did not decrease until the end of the stimulus [53]. Alice Teghil et al. [60] showed that only the timing accuracy of a duration-reproduction task with irregular stimuli was predicted by trait-like interoceptive awareness as assessed by the Self-Awareness Questionnaire. This indicates that the general ability to be more aware of somatic states helped subjects to be more accurate in time perception when no regular external signals were available. They had to rely on the dynamics of their bodily selves to judge duration; “The body functions as an internal clock for our sense of time” [61].

According to this theoretical embedding, the interoceptive awareness of the bodily self relates to the awareness of time. This, in turn, should cause an overestimation of durations, making time seem to slow down [3,91] and increasing the probability of experiencing boredom. London and Monello [104] manipulated a clock to show ten or thirty minutes having elapsed during the completion of a task, while the experiment only lasted twenty minutes in both conditions. When the clock showed that ten minutes had passed, the participants reported more boredom because they correctly thought that more time had passed. Boredom served as an emotional explanation for the time seemingly passing slowly. The feeling of boredom arises at least partly from a perceived slower passage of time [105], which may derive from an increased awareness of bodily sensations. The opposite also seems plausible; the more frequently one experiences episodes of boredom, the more one is used to being focused on their bodily self, since external stimuli do not attract one’s attentional focus.

#### 4.3.3. Correlation of Interoceptive Awareness (SAQ) with Metacognition of the Sense of Time (MQT-Self)

There was an association between interoceptive awareness and metacognition on time, particularly driven through the beliefs about what impacts one’s time perception (MQT “self” subscale). However, it should be noted that this was a rather weak correlation. A possible explanation for this relationship is that the more self-aware of bodily states related to emotions, the more aware an individual is also of how their level of arousal and their emotional state shape time perception.

### 4.4. Limitations and Directions for Future Research

The results of the analyses should be considered in light of limitations. For appropriate sample sizes in validation studies, some scholars’ blanket recommendations suggest that the higher the number of participants the better, and fair when N > 200 [106]. However, Mundfrom and coworkers [107] analyzed this question more thoroughly in simulation studies considering the variables-to-factors ratio, number of factors, and the level of communality. Inspecting their results, the necessary sample sizes for the validation of each of the questionnaires are considerably exceeded with the N of the present study. In our sample, the maximum number of items in a questionnaire was 28 (for the SAQ and the BPS, respectively) and the final sample size of N = 273 subjects is nearly ten times the amount of the items in the questionnaire. There is also a concern for the representativeness of the sample. Although the age range of the participants was large, the distribution was right-skewed, there was a majority of young participants (^2^/_3_ under 30 years old). Additionally, the sample was highly educated, with only eight participants not having a university-entrance degree. The sample consisted mainly of female participants (70%). These peculiarities might have affected the factorial structures observed; more variance would be desirable for generalization of the findings.

In the present study, we focused on an exploratory analysis of the factor structures of the German questionnaires. A major limitation of the EFA is that it is a completely data-oriented statistical method based on correlations among variables which entails that it can produce factors that are hard to interpret. This was not the case for our questionnaire structures: coherent and robust factor solutions reproduced constructs that were already known from previous studies and they subsumed homogeneous items, thus making it easy to interpret and acknowledge the shared content. The theoretical understanding of the constructs assessed by the questionnaires supported the factorial solutions. However, the structures found here require confirmation (CFA) with other samples in future studies. A more profound external validation is also sought using other psychometric measures that assess similar (convergent validity) or unrelated constructs (discriminant validity). Such correlations with other validated measures are needed to support the criterion validity of the German questionnaires presented here.

## 5. Conclusions

Four questionnaires, a three-factorial BPS-20 assessing boredom (Appendix A), the CFC-10 (and its highly negatively correlated 5-item subscales CFC-I and CFC-F) for gauging the consideration of immediate and future consequences (Appendix A), a unidimensional scale for interoceptive awareness (SAQ-17; Appendix A), and an abbreviated MQT version (Appendix A) with only ten instead of twelve items for each of the subscales (“self” and “others”) measuring metacognitive beliefs about factors influencing time perception in humans, were validated in German in the present study. The factorial structure of the MQT was hard to replicate in the EFA. Results showed that metacognition of factors influencing one’s time perception is highly similar to that of beliefs about others’ time perceptions. Three strong associations were also found that relate well to theory and prior empirical findings. The consideration of future consequences is negatively linked to boredom proneness, and interoceptive awareness of one’s bodily sensations is positively linked to boredom proneness. Interoceptive awareness was linked to metacognitive knowledge about what shapes one’s sense of time. These correlations confirm the reliability of our data and may serve as initial indicators for the external validity, although testing external validity of the questionnaires was not the explicit focus in the present study. The current analyses provide evidence for substantiating knowledge about the relationship between four constructs related to the perception of time. The German inventories can now be used for subsequent investigations but require further examination of the robustness of the factorial structure in a future confirmatory factor analysis study with another sample. A more profound external validation using other validated psychometric measures that assess similar (but not the same) constructs as the present scales is intended. Using these questionnaires, we plan to investigate how an individual’s trait-like variables influence the subjective passage of time and duration estimates of given periods in shorter (seconds and minutes) and longer (life periods) time ranges [108].

## Figures and Tables

**Figure 1 ijerph-17-08477-f001:**
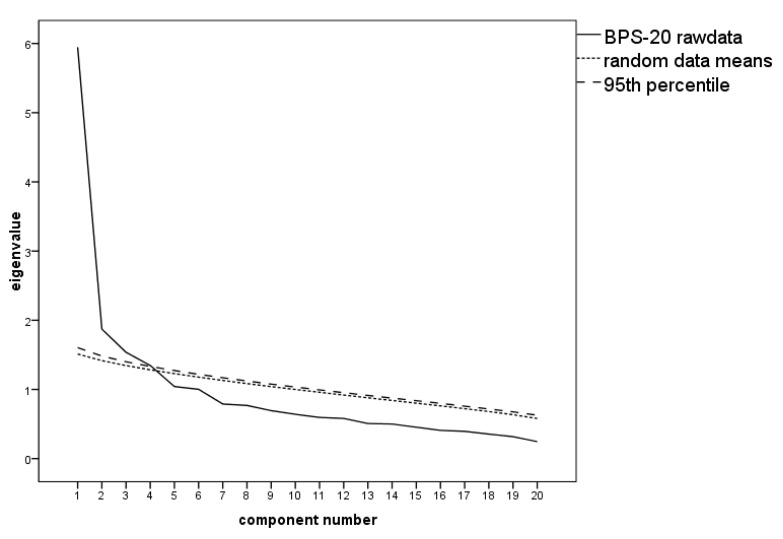
Scree plot with the parallel analysis of the Boredom Proneness Scale (BPS-20) suggesting three factors to be extracted.

**Figure 2 ijerph-17-08477-f002:**
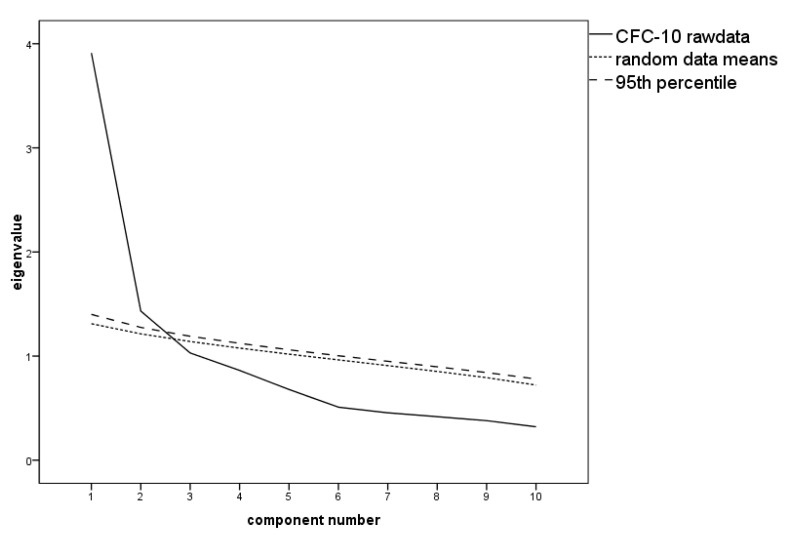
Scree plot with the parallel analysis of the Consideration of Future Consequences Scale (CFC-10) suggesting two factors to be extracted.

**Figure 3 ijerph-17-08477-f003:**
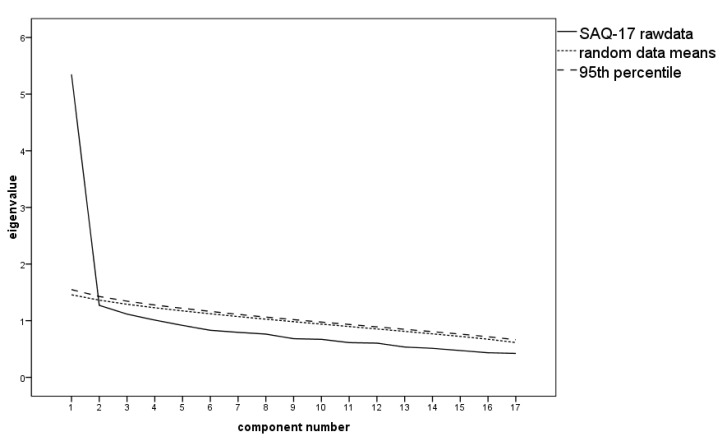
Scree plot with the parallel analysis of the Self-Awareness Questionnaire (SAQ-17) proposing extraction of one factor.

**Figure 4 ijerph-17-08477-f004:**
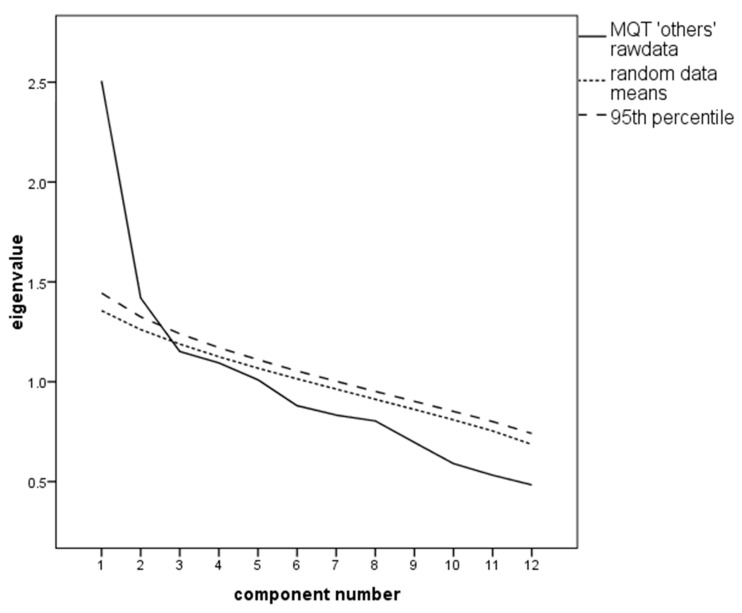
Scree plot with the parallel analysis (PA) of the Metacognitive Questionnaire on Time (MQT) “others” scale proposing extraction of two factors.

**Figure 5 ijerph-17-08477-f005:**
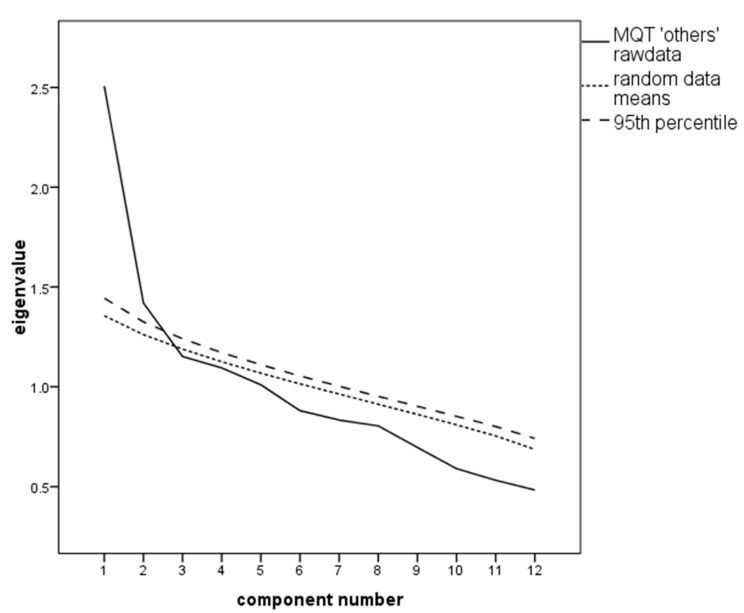
Scree plot with the PA of the MQT “self” scale proposing extraction of two factors.

**Table 1 ijerph-17-08477-t001:** Statistics of the demographic data and time on task of N = 273 participants.

Measure	Age [Years]	Educational Level	Time to Complete the Survey [s]
Mean (SD)	31.22 (12.46)	4.54 (0.56)	975.75 (438.69)
range	17–83	3–5	500–3369

**Table 2 ijerph-17-08477-t002:** Exploratory factor analyses (EFA) communalities and factor loadings after the oblique rotation of the shortened BPS-20.

Item No.	Communality (h^2^)	Factor 1 Lack of External Stimulation	Factor 2 Lack of Internal Stimulation	Factor 3 Constraint Inability to Wait
1 *	0.33	**0.49**	**−0.47**	0.15
4	0.45	**0.53**	**−0.60**	0.13
5	0.27	**0.50**	−0.31	0.19
8 *	0.28	0.33	**−0.51**	0.06
9	0.21	**0.45**	−0.24	0.12
10	0.40	**0.53**	**−0.52**	0.05
11 *	0.42	**0.52**	**−0.54**	0.25
12	0.37	**0.58**	−0.38	0.18
13 *	45	0.30	**−0.63**	0.30
14	0.60	**0.47**	**−0.74**	−0.02
15 *	0.54	0.15	−0.07	**0.73**
16	0.49	0.39	**−0.69**	0.04
17	0.37	0.12	−0.08	**0.61**
19	0.50	**0.68**	−0.12	0.17
20	0.43	**0.65**	−0.25	0.04
21	0.49	**0.69**	−0.28	0.01
23 *	0.22	0.01	**−0.43**	0.00
24 *	0.20	0.20	**−0.44**	0.11
25	0.29	**0.53**	−0.32	0.12
26	0.34	**0.56**	−0.17	0.23
**eigenvalue**		**5.95**	**1.87**	**1.54**
**% of variance**	**46.79**	**29.74**	**9.36**	**7.68**
**Cronbach’s alpha**	**0.86**	**0.84**	**0.83**	**0.64**

*Note.* * Items have been inverted before conducting the EFA. Factor loadings of λ > 0.40 are in bold.

**Table 3 ijerph-17-08477-t003:** Descriptive statistics of the BPS-20 and its subscales according to the exploratory factor analysis for the whole sample of N = 273 participants.

Measure	BPS-20 Sum Score	BPS “External” Sum Score	BPS “Internal” Sum Score	BPS “Constraint” Sum Score
Mean (SD)	59.78 (19.73)	32.26 (12.45)	26.72 (10.61)	6.29 (3.34)
range	23–133	11–72	11–60	2–14

**Table 4 ijerph-17-08477-t004:** EFA single-factor solution: communalities and factor loadings of the CFC-10 items.

Item No.	Communality (h^2^)	CFC-10 Single Factor
1	0.29	−0.54
2	0.33	−0.58
3 *	0.52	0.72
4 *	0.31	0.56
7	0.22	−0.47
9 *	0.25	0.50
10 *	0.25	0.50
11 *	0.49	0.70
13	0.26	−0.51
14	0.35	−0.59
**eigenvalue**		**3.91**
**variance**		**39.12**
**Cronbach’s alpha**		**0.83**

*Note. ** Items have been inverted before conducting the EFA. Factor loadings of λ > 0.40 are in bold.

**Table 5 ijerph-17-08477-t005:** Descriptive statistics of the CFC-10 sum score of CFC-F and inverted CFC-I items and its proposed subscales CFC-immediate and CFC-future for the whole sample of N = 273 participants.

Measure	CFC-10 (N = 273)	CFC-I Sum Score	CFC-F Sum Score
Mean (SD)	51.65 (8.97)	13.90 (5.28)	25.56 (4.81)
range	25–70	5–30	7–35

**Table 6 ijerph-17-08477-t006:** EFA communalities and factor loadings of the items in the SAQ-17 single-factor solution.

Item No.	Communality (h^2^)	SAQ-17 Single Factor
1	0.18	0.42
4	0.30	0.55
7	0.23	0.48
8	0.44	0.67
9	0.38	0.62
10	0.18	0.42
12	0.17	0.41
14	0.28	0.52
16	0.27	0.52
17	0.27	0.52
18	0.30	0.55
20	0.37	0.61
21	0.30	0.55
23	0.18	0.42
25	0.26	0.51
26	0.30	0.55
28	0.24	0.49
**eigenvalue**		**5.35**
**% of variance**		**31.47**
**Cronbach’s alpha**		**0.86**

*Note.* Factor loadings of λ > 0.40 are in bold.

**Table 7 ijerph-17-08477-t007:** EFA single-factor solution after the direct oblimin rotation of the MQT “others” items after having removed items with low communality and factor loadings.

Item No.	Communality (h^2^)	MQT “Others” Single Factor
1	0.48	0.69
17 *	0.18	0.42
18 *	0.21	0.46
23	0.37	0.61
**eigenvalue**		**1.89**
**% of variance**		**47.29**
**Cronbach’s alpha**		**0.61**

*Note.* * Items have been inverted before conducting the EFA. Factor loadings of λ > 0.40 are in bold.

**Table 8 ijerph-17-08477-t008:** Descriptive statistics of the MQT, separate for the two sum scores of the (shortened 10-item) “others” and “self” subscales and the respective “attention” and “emotion” factor sum scores for the whole sample of N = 273 participants.

Measure	MQT “Others”	“Others” Attention	“Others” Emotion	MQT “Self”	“Self” Attention	“Self” Emotion
Mean (SD)	40.23 (3.92)	28.24 (2.93)	11.99 (1.74)	40.99 (4.17)	28.96 (3.08)	12.03 (1.73)
range	30–50	21–35	6–15	28–50	19–35	7–15

**Table 9 ijerph-17-08477-t009:** EFA single-factor solution after the direct oblimin rotation of the MQT “self” items after having removed items with low communality and factor loadings.

Item No.	Communality (h^2^)	MQT “Self” Single Factor
4	0.36	0.60
14	0.45	0.67
15 *	0.17	0.41
20	0.31	0.56
21	0.18	0.43
**eigenvalue**		**2.14**
**% of variance**		**42.82**
**Cronbach’s alpha**		**0.65**

*Note.* * Items have been inverted before conducting the EFA. Factor loadings of λ > 0.40 are in bold.

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
