# Peer review of "A German Validation of Four Questionnaires Crucial to the Study of Time Perception: BPS, CFC-14, SAQ, and MQT"

_ijerph, 2020, doi:10.3390/ijerph17228477_

Round 1

Reviewer 1 Report

Please add the following sub-section on top of page 9:

3.1.1 Three-factorial EFA of the BPS

For the 28-item  BPS, the initial parallel analysis identified three factors to be extracted. Communalities h2 <.30 were observed for thirteen items ....

Author Response

Please add the following sub-section on top of page 9:

3.1.1 Three-factorial EFA of the BPS

For the 28-item  BPS, the initial parallel analysis identified three factors to be extracted. Communalities h2 <.30 were observed for thirteen items ....

Thank you very much for this helpful remark (and your review). It seems that this sub-section headline has been lost in the extensive revision process. We added it accordingly in the proposed location [line 382].

Reviewer 2 Report

Overall is an interesting article. However, the introduction is very long, and a comprehensive structure is needed to stick to the point. The description of the methods is somewhat chaotic, and I found it difficult to follow understand what the authors have done step by step. The analysis seems appropriate, but some points need clarification.
Line 211:This first German translation was then translated back into English (CFC-14; BPS), Italian (SAQ), and French (MQT). As I understand the original versions of BPS and SAQ were in Italian and French? If so, please state this in the methods section.
Line 233:... Nineteen of 232 these individuals…Do the authors mean those who were excluded?
Line 276:.....Test-retest reliability over one week... How many participants completed the retest and how they selected from the pool of 314 participants? Also, the test-retest applied only to the Boredom Proneness Scale (BPS) and not to the other scales? This should be clearly stated in the analysis method.
Line 381:.. The authors stated that Five of the items had factor loadings of less than λ = .40 and therefore they are excluded.... However, in table 2 some low communalities are presented. How the author handled this point? The authors should state explicitly if the item exclusion criteria were (a) low communalities, (b) low loadings, or both for all analyses.
Also, correlations with other validated measures are lacking. This should be discussed also
The limitations are well introduced but a discussion about the adequacy of the number of the total sample size for so many items should also be discussed. Also, correlations with other validated measures are lacking e.g., criterion validity. This should be discussed also.

Author Response

Overall is an interesting article. However, the introduction is very long, and a comprehensive structure is needed to stick to the point.

The description of the methods is somewhat chaotic, and I found it difficult to follow understand what the authors have done step by step.

The analysis seems appropriate, but some points need clarification.

Thank you very much for your review and the helpful comments. We integrated all your specific remarks in the updated version of our manuscript. For the introduction that in your opinion was lengthy, we were confronted with opposing remarks in the review process. There were reviewers who asked us to add information on the constructs and possible alternatives to the questionnaires and their uniqueness. There were reviewers who appreciated the concise but informative overview we provide. Please consider that we had to cover four constructs (relating to each of the questionnaires) and we restrained ourselves to three pages. As for the structure, we hope you find the additional short section helpful which we provide just before we present the respective constructs. In this section, we outline the structure of the following sections to facilitate the orientation [lines 51-57]. For the description of the methods, we first report the setting, procedure, and participants` characteristics. Second, we provide psychometric information on the respective questionnaires before (third point) we describe the statistical analyses. At the sections you pointed out, we disambiguated and once again inspected them to clarify our procedure in the best possible way.

Line 211:This first German translation was then translated back into English (CFC-14; BPS), Italian (SAQ), and French (MQT). As I understand the original versions of BPS and SAQ were in Italian and French? If so, please state this in the methods section.

As stated in the methods section (section 2.1), the back-translation method was used to create the German versions of the questionnaires in order to most accurately reproduce the content of the items of the original language. We slightly rephrased the sentence to clarify that the original versions of the questionnaires were English for both the CFC-14 and BPS; Italian for the SAQ; and French for the MQT. This is again stated in the methods sections on the specific questionnaires [line 220].

Line 233:... Nineteen of 232 these individuals…Do the authors mean those who were excluded?

Yes, we relate to the participants that had to be excluded from the analyses. We clarified that in the manuscript [lines 246-253].

Line 276:.....Test-retest reliability over one week... How many participants completed the retest and how they selected from the pool of 314 participants? Also, the test-retest applied only to the Boredom Proneness Scale (BPS) and not to the other scales? This should be clearly stated in the analysis method.

This mentioned retest (described in the methods section on the inventories) was not conducted in our sample. As we wrote: „Test-retest reliability over one week was found to be high in an English sample (r =0.83)“. We presented data of the original English validation [line 281-282]. In the whole manuscript we again checked and tried to state as clear as possible when we related to descriptions of the previous results of the original validation studies. The results of our validation analyses of the German versions are to be exclusively found in the results section of the manuscript.

Line 381:.. The authors stated that Five of the items had factor loadings of less than λ = .40 and therefore they are excluded.... However, in table 2 some low communalities are presented. How the author handled this point? The authors should state explicitly if the item exclusion criteria were (a) low communalities, (b) low loadings, or (c) both for all analyses.

We tried to state this explicitly in the methods section: „Items were removed from the German versions of the questionnaire when their communality was below h2 < .30 and the factor loadings were smaller than λ = .40 on all factors [78].“ That is, when both exclusion criteria were met, items were removed (Leech et al., 2012). This explains why some items with communalities < .30 are retained and presented in the table as long as they loaded sufficiently high on the respective factors (λ .40). We tried to disambiguate this in the manuscript [lines 351-353].

Also, correlations with other validated measures are lacking. This should be discussed also

You are right about the lack of external validation with other validated measures in the present study for evidence of criterion validity. In the manuscript, we tried to point this out several times (see the abstract, limitations, conclusion): „A more profound external validation is also sought using other psychometric measures that assess similar (convergent validity) or unrelated constructs (discriminant validity).“ Nevertheless, we added another sentence to the limitations section in the discussion to highlight this lack of external validation in the present study [lines 808-811].

The limitations are well introduced but a discussion about the adequacy of the number of the total sample size for so many items should also be discussed.

This is an important remark as the quality of validation studies depends on the sample size, suggesting that the higher the number of participants the better. However, Mundfrom et al. (2005) analyzed this question thoroughly in simulation studies considering the variables-to-factors ratio, number of factors, and the level of communality. Inspecting table 1 of the Mundfrom et al. (2005) article, the necessary sample sizes for the validation of each of the questionnaires are considerably exceeded with the N of the present study. In our sample, the maximum number of items in a questionnaire was 28 (for the SAQ and the BPS, respectively) and the final N = 273 is nearly ten times the amount of items. We added a paragraph to the limitations section to include discussion on this topic [lines 786-794].

Also, correlations with other validated measures are lacking e.g., criterion validity. This should be discussed also.

We think this remark refers to the remark on external validity above to which we responded.

Round 2

Reviewer 2 Report

I would like to compliment the authors for their work. No further remarks.

This manuscript is a resubmission of an earlier submission. The following is a list of the peer review reports and author responses from that submission.

Round 1

Reviewer 1 Report

The topic of the paper is appropriate for the journal and could add to the existing state of science. The main problem is (still) the small sample size and that the study participants are not representative. In addition, the analyses are straightforward but lengthy and not very attractive to read. I am wondering whether the authors understood the feedback I provided to the last version they submitted and why they did not apply my feedback more adequately to improve their manuscript. In specific, I see the need to aggregate the analyses and to do them more advanced with, e.g., SEM. This would be the precondition to increase the readability and make the manuscript more interesting to read. Only afterwards, the small problems, which I will outline nonetheless in the following, make sense to work on! In general, there are many grammatical and typographical errors that should be modified.

Abstract

In the abstract no aim and conclusion are described. Adding one or two sentences explaining the aim of the study allows the reader to understand the context in which the research is being done. Additionally, the author should add a brief sentence summarizing the conclusion of the study. The keywords depict the main idea of the study. 

Typographical & Grammatical errors – Abstract 

Line 11- “14 Scale” should be changed to “14 scales,” as the singular noun “scale” follows a number other than one. Consider changing the noun to the plural form. 

Line 12 - … and Self-Awareness Questionnaire (SAQ)) using a back-translation method. , Exploratory … Remove the extra comma.

Line 18- Delete “own,” it is redundant.

Line 20- Consider changing “in reference to” to “in regard to.”

Introduction and aim

The aim is not clearly stated, nonetheless it could be drawn upon. Hence, it is suggested that it is elaborated further to follow logically the literature review. The following points discuss improvements that could be used for the optimal flow of the text: 

  • Are lines 29-33 the aim of the study? If so, state it clearly.
  • Line 34 & 35- “Individuals are more likely to consider future consequences when the future event feels less distant.” o Is the academic reference for such statement the same one for lines 35-38? If not, the author should specify where are they drawing these conclusions from.
  • Line 39- “Basically all the construct are linked to time perception.” Although scientific references were previously provided, such statement lacks scientific argumentation and requires better phrasing.

Such statement can be phrased in the following way: 

  • “ As XX indicate, it can be concluded that all the constructs are linked to time perception.”
  • The following notes can be omitted, but can be added to enhance the introduction:
    • Line 41- What translated scales, be more specific.
    • Line 42- What correlations?
  • Line 71- “The German word…” sentence is an interesting fact that resembles the main idea of the text, nonetheless, it is misplaced as it presents a separate idea that could be connected before to ensure optimal flow.
  • Lines 73-85 – The author notes different boredom spheres, such as the professional and educational contexts. Consider classifying lines 73-78 as a social or personal context to be more specific.
  • Line 78- what exactly is referred by “…rather than the other way around.” Elaborate more on it.
  • Line 94-95- The context specific trait-boredom concern the SSS scale? Set further context regarding line 96, it is not clear why the word “however” is used.
  • Line 106-107- Relevant idea, nonetheless, it is a hard-to-read sentence. Try rephrasing such statement in an efficient yet understanding manner: “…(e.g., performing well in exams [35]) involve self-control and not giving in to the alluring, but, in the long run, hindering, hedonistic activities in the present.” 1. There is a parenthesis missing to close the sentence. 2. involves not involve. 3. the sentence is slightly lacking in clarity, maybe remove the comma after hindering or change it to “hinders hedonistic activities in the present”? I have read the sentence multiple times and something seems to be missing to clarify understanding.
  • Line 161- The “precursor” questionnaire is the one designed by Longarzo?
  • Line 181- What do you refer by “above points”? Be more specific. Such statement requires further explanation.

Typographical & Grammatical errors – Introduction

Line 29- Change to “on the basis of” to “based on.”

Line 30- Remove redundancy, delete “own.” 

Line 39- Remove “Basically” it is unnecessary in this sentence.

Line 40- Add commas “we ,therefore,…”

Line 43- Change the wording “with reference to” to “concerning/regarding.”

Line 51- Add comma after “disengagement.” 

Line 54- Rephrase “This last-mentioned…”

Line 56- Parenthesis not allocated correctly, delete “and” or delete the parenthesis.  

Line 57- “…when persons want” rephrase, change to “people.”

Line 68- “ Supporting this perspective of a …” Rephrase, sentence could be more specific. Make sure the two clauses use the correct verb tenses. 

Line 72- Remove the redundancy/tautology, “period of time.” 

Line 77- The form of the noun “psychopathology” does not appear to be correct when used before the phrase “such as.” Consider changing it to

“psychopathologies.”

Line 82- “a proneness,” delete the “a.”

Line 93- The noun phrase “10-item subscale” seems to be missing a determiner before it. Consider adding an article or “a”. 

Line 97- “externalising” is a non-American variant, watch out for consistency throughout the text: here, externalizing not externalising.

Line 98- Add hyphen to “sensation seeking.” Line 103- Add hyphen “risk taking.” Line 104-105- Add respective commas. 

Line 109- “in the here and now,” change to “present.”

Line 111- “a stronger,” remove “a” since it may be redundant when used with the uncountable noun orientation.  Line 112- Change to “that promote.”

Line 116- Conciseness- Consider changing “there are also people who focus” to

“Some people focus.”

Line 122- Watch out for verb agreement in the two clauses. 

Line 129- Change “It is notable” to “Notably,”. 

Line 145- delete “basically”

Line 163- delete “actually”

Line 166- Delete the comma after “cues.”

Line 174- Add the respective commas to “however”: “The eight subscales and the items however are not …” Comma before and after however.

Line 188- Add “the” to “enjoyment”

Line 191- Rephrase, it is a hard-to-read sentence. 

Materials and methods

It should be elaborated further on how the respondents were recruited (i.e. was the survey promoted on social media channels?)

It is stated that ethical approval was sought, nonetheless, it could be elaborated further on the explanation on why it was sought and the ethical objections to the study. The following points highlight details that could be further discussed:  

  • 2. Procedure- Line 214- Provide further description of what happened to each participant. Include the online survey software, how did they access such software (email?) or just demographic information? Include overall response rate.
  • 3 Participants
  • Lines 227-234- What was the benchmark for comparing the time spent on each questionnaire?
  • Line 228 - … spending less than 5s spent on … Remove “spending” or “spent”.
  • Line 233- Table 1 lists the demographic data of the remaining N = 273 (187 female, 85 male, 1 diverse) participants. The position of this sentence in the paragraph does not appear logical, perhaps in the beginning would be better or if it is rephrased
  • Line 240- only one admitted having “only” advanced knowledge of German: The first Only can be omitted (or kept depending on intended tone) as it is repeated twice (with quotation marks, for the second time). Perhaps another word can be used instead of the second “Only”
  • Line 267, 268 - Note that we changed the poles of the scale to avoid accidental wrong answers because a higher level was directed to the right, higher number on the other questionnaires. A word might be missing, or should be added. The sentence does make sense when adding that last clause the way it is.
  • Line 245 should be moved to this part, as it provides the reader with the average time each respondent was expected to complete the questionnaire in. Furthermore, it would set a clearer context for line 231 and the “exaggerated” time.
  • Line 457- Including age as a possible confounding variable * in: Too wide a space between “variable” and “in” – Typing error
  • Line 469 - ​Descripitve statistics of the CFC-10 sum score of CFC-F and inversed CFC-I items … Descriptive is the correct spelling. 2. As stated above, preferable to use “inverted” instead of “inversed”.
  • Line 302- Why 0-112? Why 112? Elaborate further on why this score was chosen.
  • Line 307- Why did the allocations of the two factors differed from the Italian version?
  • Line 317, 318 - However, in this second study, the ‘others’ scale suggested that the knowledge or estimations of others’ temporal distortions is not reliable … “Is not” should be changed to “are not” because the word “or” is used with a singular word (knowledge) and a plural word (estimations).
  • Line 315-318- What was is the second study about? How is the topic research of it relevant to your study? Why do you use this study and not others? Consider adding the authors or adding more background about it.

Typographical & Grammatical errors – Materials & Methods 

Line 205- Conciseness- Change “the aim of the study was” to “The study aimed.”

Line 314- “The reliability”

Line 318- “…temporal distortions are not reliable.”

Line 325- “ …which does not…” Watch out for verb agreement. The next sentence uses past tense and consequently, line 328 suddenly changes to present.  Line 329- Change “in accordance with” to “following.”

Line 340- Commonality or communality 

Line 341- Watch out for the word “oblimin,” despite it being a statistical term, it is not recognized as a word.  Line 351- “that lead”

Line 354- “the deletion”

Results

  • Line 393- Provide further explanation on why items 1,10, and 11 were allocated to both factors.
  • Line 500- How did you “sorted it out”? Provide further details. 

Typographical & Grammatical errors – Results 

Line 367- “Test” should be in lowercase letters. 

Line 369- “Screeplot” should be written as “screen plot” also in lines 379, 422, 496, 518, and 553

Line 383 - Note. * Items items have been inversed before conducting the EFA. The word “items” is used twice. 

Line 416- “Two-factor”

Line 418 - … the factor loadings were or positive or negative for both factors. There is an extra “or” here, it should be removed.

Line 423- “Single-factor”

Line 427 - ​. After again removing items… ​There is an extra period at the beginning of the sentence.

Line 481- Change “over” to “of.”

Line 483 - ​KMO was very good, KMO = .89; the Bartlett​ ​test significant… ​The word “was” is missing between Bartlett and test. 

Line 484- *the* variance 

Line 489 - ​This is also in line with the observation that also in the Italian and English versions … ​There are 2 also’s here which make the sentence redundant; “that also in the” is a bit heavy on the tongue. I suggest keeping the first “also” and correcting “that also in the” to simply “​in the​” and remove “that also”.

Line 497- “Single-factor”

Line 511- Rephrase- “Hence, we…”

Line 524- *The* extraction

Line 525- *the* deletion

Line 529- derive

Line 561- *the* maximization

Discussion

The discussion needs to be adapted to all changes with the methods and results (see my general feedback in the beginning) and further reference can be made to important points discussed in the introduction such as the context specific trait-boredom. It is recommended to use more scientific evidence to the support the noted claims. The following points highlight details that could be further discussed:  

  • Line 661- What would happen if the MQT was analyzed together in two PAs and not separately?
  • Line 664 to 666 - ​Their content was exactly the same one time phrased for the assessment of metacognitive beliefs about the others’ sense of time, one time for the own time perception. ​ I think there should be a comma after “the same” or somewhere else in the sentence, otherwise the sentence doesn’t make much sense. 2. Change “the” to “their” own time perception.
  • Line 667- “These might therefore reflect items which worked very well for the German sample in terms of interpretation and rating.” o Is this a potential limitation?
  • Line 677- 679- Did focusing only on internal consistencies posed a limitation to the study?
  • Line 688 - ​… separate between the evaluations of the own and others’ time perception. ​“Their” should replace “the” for the sentence to make sense.
  • Line 706 - ​It remains speculative why​ ​at least for the BPS and the SAQ​ ​there… ​Add a comma after “why” and after “SAQ”, to make the sentence more readable.
  • Line 708-709- Consider adding a citation or providing an academic reference to support your claim.
  • Line 717- Provides specific evidence regarding why 8 elements were deleted. It would also be useful if academic references were used to support the claims.
  • Line 719 - ​Considering the content of these items​ ​it does not seem probable​ … Add a comma after “items”.
  • Line 741- Elaborate more on the “theoretical understanding of the constructs.”
  • Line 747- Provide specific evidence to support your claim.
  • Line 759- Good point to make reference to your introduction- line 34,35,39, regarding time perception.
  • Did you find any findings regarding the contexts that were included in the introduction (line 73-85)?

Typographical & Grammatical errors – Discussion 

Line 618- Delete “definitely.”

Line 619- Delete “clearly.”

Line 622- Add comma after 2016

Line 624- Delete “clearly”

Line 634- Change relation to relationship

Line 641- Change seems to seem 

Line 666- Add respective commas to “therefore” and change “which worked” to “that worked”.

Line 676- Change to “that have”

Line 677-679- Watch out for verb tense agreement. 

Line 689- Change “of” to “between.”

Line 701- Change “with the aim to” to “to”

Line 727- Change “for” to ‘to.”

Line 731- Add “the” to deletion. 

Line 734- Delete “the” from reliability.

Line 736- Delete clearly 

Line 778- Change “on” to “of.”

Line 779- Remove “own.”

Line 786-787- Sentences could be organized more smoothly to ensure optimal flow. Consider reorganizing.  Line 789- Add “the” to awareness. 

Line 789 - This, in turn, should cause an overestimation of durations, making time to seem to slow down … Removing the “to” will make the phrase much better to read.

Line 792- Delete “actually.”

Line 797 to 798 - ​… the more frequently I experience episodes of boredom, the more I am used to being focused on my bodily self, since external stimuli do not attract my attentional focus. ​I don’t know if it is acceptable, but maybe switch out referring to yourself as “I” and give a more broad example with “one”or another pronoun: ​the more frequently one experience episodes of boredom, the more one is used to being focused on their bodily self, …

Line 801- Delete “own.”

Line 804 - ​… how the own level of arousal and the emotional state shape time perception.

Change “the” to “their” for the sentence to make more sense.

Line 808- Delete “that is.”

Line 817 - ​homogenous items, thus making it easy to interpret and acknowledge the shared content. ​Homogeneous is the correct spelling.

Line 821- “that assess.”

Conclusion

Line 830- Delete “own.”

Line 833- Delete “own.”

Line 834- Delete “the” from metacognitive, delete “own.” Line 836- Add “the” to external. 

Line 837 - ​… in the present study. The present study provides evidence …​ “the present study” is used twice, right after one another. I suggest changing one of the “present” into a synonym, for example “current” to make it more readable. 

Line 841- “that assess.”

Line 844- Remove tautology, change “period of time” to “periods.”

Line 875 - ​Boredom in achievement settings: Exploring control-value antecedants and performance outcomes of a neglected emotion. ​Antecedents with an “e” is the correct spelling.

Reviewer 2 Report

There are still typos in the revised version and the authors need to fix these typos before the publication of the manuscript. 

Reviewer 3 Report

I am satisfied with the authors' revisions to their manuscript based on reviewer comments. I have a couple of small remaining recommendations:

Table 2: now that the loadings are bolded over .4, I'd simply add a note stating what the bolding means for the reader.

I still do not see the value of the section on education. The R2 values are quite small. More importantly, within the context of psychometric validations, having differences in mean scores on a measure by a grouping variable is only relevant if there is measurement noninvariance across educational level (i.e., differences in factors/loadings/intercepts etc. In other words, differences in educational level have no bearing on your study unless you're testing for invariance (but you're not here). Accordingly I'd remove section 3.2.3 altogether.